# Biochemistry of the Endocrine Heart

**DOI:** 10.3390/biology11070971

**Published:** 2022-06-27

**Authors:** Jens P. Goetze, Emil D. Bartels, Theodor W. Shalmi, Lilian Andraud-Dang, Jens F. Rehfeld

**Affiliations:** Department of Clinical Biochemistry, Rigshospitalet, University of Copenhagen, 9 Blegdamsvej, DK-2100 Copenhagen, Denmark; emil.daniel.bartels@regionh.dk (E.D.B.); tfn714@alumni.ku.dk (T.W.S.); xqd367@alumni.ku.dk (L.A.-D.); jens.frederik.rehfeld@regionh.dk (J.F.R.)

**Keywords:** ANP, BNP, CNP, cholecystokinin, PAM, natriuretic peptide

## Abstract

**Simple Summary:**

Besides being a muscle and an electrochemically active organ, the heart is a true endocrine organ. As endocrine cells, cardiac myocytes possess all the needed chemical necessities for translation, post-translational modifications, and complex peptide proteolysis. In addition, intracellular granules in the cells contain not only peptides destined for secretion but also important granin molecules involved in maintaining a regulated secretory pathway. In this review, we highlight the biochemical phenotype of the endocrine heart, recapitulating that the cardiac myocytes are truly and fully capable endocrine cells.

**Abstract:**

Production and release of natriuretic peptides and other vasoactive peptides are tightly regulated in mammalian physiology and involved in cardiovascular homeostasis. As endocrine cells, the cardiac myocytes seem to possess almost all known chemical necessities for translation, post-translational modifications, and complex peptide proteolysis. In several ways, intracellular granules in the cells contain not only peptides destined for secretion but also important granin molecules involved in maintaining a regulated secretory pathway. In this review, we will highlight the biochemical phenotype of the endocrine heart recapitulating that the cardiac myocytes are capable endocrine cells. Understanding the basal biochemistry of the endocrine heart in producing and secreting peptides to circulation could lead to new discoveries concerning known peptide products as well as hitherto unidentified cardiac peptide products. In perspective, studies on natriuretic peptides in the heart have shown that the post-translational phase of gene expression is not only relevant for human physiology but may prove implicated also in the development and, perhaps one day, cure of human cardiovascular disease.

## 1. Introduction

The endocrine heart is a relatively new discovery. From the 1960s, granules observed by electron microscopy in cardiac myocytes led to a hypothesis of an endocrine heart that could synthesize peptide hormones. The first true endocrine evidence was reported in 1981, where Adolfo de Bold and his co-workers reported on an atrial natriuretic factor (ANF) that potently stimulates renal natriuresis [1]. From that observation, molecular identification of the factor atrial natriuretic peptide (ANP), and two structurally related peptides, B-type natriuretic peptide (BNP) and C-type natriuretic peptide (CNP), were identified [2,3,4,5]. ANP and BNP are both expressed in cardiomyocytes and are true cardiac hormones. CNP, on the other hand, is mainly a local regulator expressed in endothelial cells, chondrocytes and the reproductive system. Although the early research on peptide biochemistry and biology suggested a simple post-translational maturation of both propeptides, we know today that the endocrine cardiomyocytes are fully capable endocrine cells with an elaborate and variable processing of peptides destined for release. Moreover, the endocrine heart also produces other peptides that may play hitherto unappreciated roles in cardioendocrine regulation of biological processes.

The endocrine heart rapidly attracted clinical attention. Measurement of ANP, BNP and their molecular precursors in plasma proved useful in diagnosing heart failure [6] (for reviews, see [7,8,9]). This diagnostic utility has been extensively studied in respect to most cardiac diseases, and although they contain prognostic value for most diseases, they remain pathogenetically central in heart failure in all its forms. As measurement of natriuretic peptides was pursued, the interest for the endocrine phenotype of the heart also increased. One major finding was the discovery of an endoproteolytical cardiac-processing enzyme, Corin, which seems critical for maturing propeptides to the active hormones [10,11]. In addition, cardiac peptides are variably O-glycosylated, which is a key regulatory step for the maturation (or lack of same) towards bioactive peptides. In this mini-review, we highlight the biochemical phenotype of the endocrine heart with a peptide-oriented narrative on the synthetic path from mRNA translation to cellular release.

### 1.1. Granules

At first glance, cardiomyocytes do not appear to be traditional endocrine cells. Their phenotype is dominated by the striated muscular orientation together with their connections (gap junctions) to other cardiomyocytes. Moreover, the nuclei of the cells are not located at one end of the cells but rather orientated as a function of the muscular arrangement in a polyploid manner. With the development of the electron microscopy, granules within atrial cardiomyocytes could also be visualized [1,12]. We now know that the granules are part of a regulated pathway towards fusion with the cell membrane and release of its contents to blood (Figure 1). As other granules, cardiac granules contain “granins”, which is a class of chaperone proteins involved in the formation of granules as well as maintaining a particular intragranular (acidic) chemical environment. Two such granins in the cardiomyocyte granules are chromogranin A and chromogranin B [13,14,15]. Both granins seem not only present but of importance for normal maturation and transportation of natriuretic peptides. Another granin is the bifunctional protein peptidyl-alfa-amidating monooxygenase (PAM), which is abundantly present in the mammalian heart [16,17]. PAM is both a granin-like protein and a catalytic enzyme for amidation of peptides. Removal of PAM from cardiomyocytes results in the removal of granules, which underscores its key role in granular formation and biology [18,19,20]. For the enzymatic function of the protein; however, no amidated peptide has yet been identified in the granules (vide infra).

As mentioned, granules are present in atrial cardiomyocytes. Ventricular myocytes, on the other hand, do not contain granules, at least not in ventricular myocytes from healthy hearts [21,22]. When ventricular myocytes are subjected to stress; however, their phenotype changes towards an atrial phenotype and begin to contain granules [23]. Our current knowledge of granules in the ventricular cells is still limited, but it is possible that the path from peptide synthesis to secretion follows a constitutive-like rather than a strictly regulated pathway. Some suggestions for storage of peptides in “normal” ventricular myocytes have been reported, where ventricular tissue, for instance, can store proforms of cholecystokinin (CCK), which is a gut hormone also expressed in the mammalian heart [24,25]. Notably, ventricular cells are notoriously difficult to culture in vitro which has excluded detailed molecular studies of peptide storage and release. Our current understanding of endocrine ventricular cells is; therefore, fragmentarily based on whole organ studies and adynamic histology.

### 1.2. Translation

A key event in translation of mRNA to propeptide is the removal of the signal peptide. The enzyme signalase cleaves an N-terminal fragment from the preprostructure while the translation occurs, and the signal peptide is supposedly then degraded. Signal peptides are often said to be molecular guidewires that ensure the ribosomal translation into the Golgi network. For the cardiac peptides, the preprostructures are thus non-existent as independent structures. In contrast to the general concept that signal peptides are degraded, signal peptides from the natriuretic peptides are expressed and identifiable [26,27,28]. Moreover, fragments of the signal peptides are even released to circulation and can be quantitated in plasma. As to how these peptides leave the cardiomyocytes is still not resolved, but clinical data suggests that their release can be rapid and precede release from cellular apoptosis and membrane lysis. As signal peptides generally are highly hydrophobic, it may be speculated that they are “carried” by another peptide or protein, perhaps by a granin.

### 1.3. O-Glycosylation

One of the early post-translational events in propeptide maturation is O-glycosylation. This complex modification involves adding a sugar-based moiety on the free oxygen atom of serine and/or threonine residues within the propeptide. Although it was earlier believed that O-glycosylation was mainly a feature on larger proteins, we now know that peptide hormones are also glycosylated. For the cardiac natriuretic peptides, it was long speculated that the prostructures were modified, as their biochemical behaviour on for instance chromatographic elution indicated the presence of forms larger than the primary sequence indicated [29,30,31]. Biochemical deglycosylation of endogenous proBNP then showed that the molecular size could be reduced to the calculated primary mass, hence strongly suggesting O-glycosylation [32]. From there, the identification of glycosylation on proANP, proBNP and proCNP has been ongoing [33,34,35,36]. One key feature of this modification is the variable glycosylation of residues close to endoproteolytic cleavage sites. One site seems to be a regulator of maturation of the bioactive hormone, where glycosylation blocks for the endoprotease and prevents maturation and subsequent bioactivity [37]. One other discovery stemming from the identification of glycosylated proforms is O-glycosylation on the actual bioactive, e.g., receptor binding hormones [33]. This modification is particularly interesting, as some animal-based data suggests that the glycosylated peptides still retain bioactivity but are less prone to degradation (Figure 2). If this holds true also in clinical studies, the hormones are released in both fast- and slow-acting forms, which is otherwise best known from pharmacotherapy, e.g., insulin substitution. Additionally, the endocrine facet of peptide release may even be regulated in terms of either fast- or slow-acting forms according to the degree of pathology.

### 1.4. Endoproteolysis

With the identification of the genes encoding cardiac natriuretic peptides, it was clear that endoproteolysis was involved in the post-translational phase of peptide expression. All three natriuretic peptides constitute the C-terminal part of the prostructures, and the common site for endoproteolysis in proBNP and proCNP is an “Arg-X-X-Arg” motif located N-terminally to the identified hormones. The prohormone convertase involved in this cleavage is Furin, which is a serine protease expressed in most cells [10,11]. ProANP, on the other hand, is cleaved by a protease located in cardiomyocytes, e.g., Corin. Corin is located within the cell membrane, which suggests that this cleavage is a late event in the regulated pathway, perhaps even a cleavage that takes place during granular fusion with the cell membrane. Corin is also present in circulation [38], which may cause cleavage and final maturation of the pro-natriuretic peptide after cellular release. Finally, it should be recapitulated that the mature natriuretic peptides are present in granules [1,2,3], which entails that transmembrane cleavage may not be the only biochemical path to bioactive ANP [39]. Other prohormone convertases are also present in cardiac myocytes. Prohormone convertase 1/3 as well as prohormone convertase 2 have been reported in cardiac myocytes [40,41,42]. Taken together, cardiac granules are capable of complex endoproteolytic maturation of its propeptide contents, and it seems reasonable to suggest that yet unidentified processing intermediate fragments stemming from the prostructures exist and may be secreted (Figure 3).

### 1.5. Other Modifications

Post-translational modifications also include amino acid modifications, e.g., phosphorylation, sulfation, acetylation and C-terminal amidation. None of these modifications have been reported in terms of cardiac natriuretic peptides. However, all modifications seem possible in the endocrine heart. For the O-sulfation ability, the enzymes responsible are the tyrosylprotein sulfotransferases, both of which are present in cardiac tissue [43]. Moreover, sulfation of granular content does take place. We reported that cardiac myocytes express the CCK gene and the resulting peptide is sulphated on 3 tyrosyl residues [24]. As the addition of a sulphate group to a peptide represents only a smaller change in molecular mass (40 Dalton), it should not be ruled out that the prostructures to natriuretic peptides in fact could contain sulphated tyrosyl residues. In parallel with sulphated proCCK in the heart, cardiac chromogranin A is also a sulphated protein. For phosphorylation, an early report suggested that phosphorylation on bioactive ANP may be an important regulator of receptor binding and downstream signalling [44].

The possibility of amidation of the peptide contents in cardiac granules still remains largely a mystery. As mentioned, the amidation enzyme PAM is abundantly expressed in the atrial granules, yet only two amidated peptides in the endocrine heart have been identified, CCK and adrenomedullin [24]. For adrenomedullin, the expression is involved in cardiac development as well as in heart failure, and the expression pattern seem to fit the regional expression of PAM, that is mainly in the cardiomyocytes in the atria. A key feature in the PAM-related enzymatic process is a peptide motif involving a C-terminal glycine residue as amide donor. Although the natriuretic peptides as we know them today are not relevant structures for amidation, it may still be that fragments from other granular propeptides and proteins could be amidated. As purification alone on the amidation feature is very difficult, it should be kept in mind that peptides identified in the granules may also contain this modification, and that such modification often entails bioactivity as a secreted peptide.

## 2. Secretion

Cardiac granules fuse with the cell membrane and release their contents into circulation. Given the regulated manner of cellular secretion, granules are formed early in the secretory pathway from the Golgi apparatus. Whether the granules are formed and then packed with peptides or whether the immature propeptides aggregate and then “bud” off the Golgi network is not fully resolved. However, proANP as a propeptide structure can aggregate, a feature that requires calcium [45,46]. Moreover, deletion of the C-terminal part of proANP does not affect granular formation, but removal of the N-terminal part abolishes granular formation. This phenotype was also observed in mice with the complete ANP gene deleted [47]. Of interest, a lack of granules has also been observed in cardiac cells devoid of PAM, which underscores the “granin” role of PAM in the heart [19]. To our knowledge, chromogranin A and chromogranin B have not been selectively silenced in cardiac cells, and their role in granular formation in the heart is thus not clarified.

Granules reaching the cell membrane are fused upon extra- or intracellular signalling. The traditional view on cardiac secretion stimulus is based on mechanical stretch, sometimes referred to as strain, of the cardiac myocytes. In physiology, this makes teleological sense as the cardiac chambers inherently are located at the centre of the circulatory system. Thus, changes in preload and afterload will be immediately registered by the cardiac myocytes, which can then respond with increased contractility (inotropy), increased heart rhythm (chronotropy) and release of potent peptides that act on the circulatory system (here named endotropy). Three mechanisms are involved in stimulated secretion of natriuretic peptides; primarily via stretch-mediated stimulation of G-coupled receptors as well as by various secretagogues, e.g., angiotensin and endothelin [9]. Later, cytokines have been shown also to stimulate secretion, possibly via activation of intracellular p38. Notably, the latter stimulation mechanism seems to be related only to secretion of BNP-containing granules, which has led to a suggestion for BNP in circulation as a specific marker in cardiac transplantation and organ rejection [48,49].

## 3. Natriuretic Peptides

The best-known hormonal products from the heart are the natriuretic peptides [1,9]. ANP and BNP are vasodilatory peptides involved in volume and fluid homeostasis and counteract the effects of the sympathetic nervous system together with the renin-angiotensin-aldosterone axis. Decreased production and release of the peptides is accordingly associated with a hypertensive phenotype and an increased risk of cardiovascular disease [50]. Decreased production and release of the natriuretic peptides has thus been named a state of natriuretic peptide deficiency and seems even involved in metabolic disturbances as noted in type 2 diabetes. As endocrine peptides from the heart, their expression and translation in cardiomyocytes has been extensively studied [9]. Nevertheless, new discoveries are still being reported on their complex biosynthesis. Although basal research on the matter was mostly performed in the early phases of the peptide discovery, the post-translational phase of gene expression is still being explored. As the peptides are not only hormones but also useful biomarkers in cardiovascular disease, the interest in their maturation has been driven by a clinical need for improving the methods for detecting the peptides and their fragments in plasma. With the current methodology, measurement of natriuretic peptides and fragments from the precursors are today very useful in excluding a heart failure diagnosis, whereas increased concentrations are indicative of—but not inclusive of—a heart failure diagnosis. As the cardiac production of natriuretic peptides increases in states with increased cardiac pre- and afterload, the post-translational phase also shifts from production of mature hormones to release of less processed—and less potent—precursor forms. In fact, this shift in disease may lead to a paradoxical state of apparent natriuretic peptide deficiency, as the concentrations of the proforms in circulation are high but the bioactivity of the system is in fact decreased [51].

Today, natriuretic peptides can be used as pharmacotherapy in heart failure [52]. However, the success rate for this therapy form is presently not high. Given the complex endogenous maturation, it seems timely to speculate that the peptides may rather hold a place in the pharmacological armoury against, for instance, hypertension associated with the metabolic syndrome. Moreover, further insights into the post-translational phase may lead to new molecular targets for intervention in human disease, as replacement therapy could, perhaps, be exchanged with therapy that leads to a more processed natriuretic peptide release from the heart. One example is the key endoprotease Corin that releases the bioactive, C-terminal peptide hormones from the precursors. By pursuing increased Corin expression and activity in the heart, endogenous peptides could alleviate hypertension and other types of cardiovascular disease. Other processing features may be targeted, and it should be expected that other endoproteases could be brought into play. Likewise, if the molecular apparatus for glycosylation in the heart can be modulated, endogenous natriuretic peptides can be altered in terms of bioactivity and pharmacokinetics. The latter is an important feature in peptide pharmacology, as natriuretic peptides, along with many other peptide hormones, have a short half-life in circulation. Long-acting peptides may be preferred if they are to alleviate human disease and still maintain a high degree of safety in pharmacotherapy.

## 4. Other Peptides

Cardiac myocytes express other peptides. One such peptide is apelin [53,54]. Apelin is a potent vasodilator and is expressed in several tissues, including the heart. Apelin research has; however, been severely hampered by a lack of reliable methods for peptide measurement. In fact, clinical studies on apelin in plasma seem to point in different directions, as both increased and decreased plasma concentrations have been reported in human cardiovascular disease [55,56,57]. Moreover, the post-translational phase of cardiac apelin is completely unknown. Cardiac release in apelin has been demonstrated by regional sampling of blood across the heart [58,59]. Where apelin peptides are located within the endocrine heart (regulated or constitutive pathway) is still not resolved, and it remains an open question how the peptides might be modified. This is not trivial as the cardiac myocytes possess the biosynthetic apparatus for most known modifications. The facet of cardiac apelin processing may in fact be the most relevant question for future research to explore. Although the apelin gene is expressed in other tissues, cardiac peptide processing may lead to cardio-specific apelin forms, which again can both be involved in cardio-specific endocrinology and serve as a cardio-specific marker in circulation. To understand the cardiac apelin system, the task at hand is to develop reliable immunoassays for detection. Then, biochemical characterization of cardiac apelin needs to be performed while keeping the lessons learned from the natriuretic peptides in mind.

The relaxins constitute a family of structurally-related peptides that were first discovered in reproductive endocrinology [60]. Ligaments in the pelvic region soften and expand close to delivery in female animals, hence the name. As a peptide structure, relaxin resembles that of insulin in that it is composed of two chains (A and B) stemming from the same precursor. In addition, the class of relaxin genes has now been established to include seven members where all have distinct expression profiles in mammals, including the heart [61,62]. Secretion of relaxin from the heart; however, does not seem to be a major source of circulating relaxin [63]. Cardiac relaxin could, on the other hand, still be a local factor in protecting against myocardial injury. This has been reported for ischemia [64] and congestive heart failure [65]. A more recent clinical trial thus aimed at using synthetic relaxin as a cardioprotective drug in human heart failure, but with negative results [66].

How relaxin is stored in the endocrine heart is still not clarified, nor is the cardio-specific processing of relaxin. The latter is interesting in the context of the endocrine heart. Prorelaxin requires several endoproteolytical cleavages before the receptor-binding peptide is formed, which could be used to identify the proteases at play. Selective knockdown of endoproteases in the cardiomyocyte followed by detailed molecular characterization would allow for a better understanding of post-translational processing beyond natriuretic peptides. Likewise, it would be relevant to examine whether relaxin is stored and follows a regulatory secretory pathway or a constitutive-like secretion. The clinical studies from measuring relaxin in circulation from patients with cardiovascular disease or trying to stimulate cardiomyocytes by exogenous relaxin could, in turn, depend on the cardio-specific relaxin maturation rather than assuming that cardiac relaxin is the same as relaxin expressed in other organs.

Finally, gut peptides can be expressed by cardiomyocytes. For instance, according to a case report, a primary tumour derived from cardiomyocytes produced a gastrin peptide, where gastrin is mainly expressed in the stomach and regulates acid secretion [67]. Moreover, the patient suffered from the Zollinger-Ellison syndrome, which is a neuroendocrine tumour syndrome seen in gastrinomas. The patient with the cardiac tumour suffered from this syndrome by severe duodenal ulcers, and the localization of the tumour was only proven post-mortem. Since then, others have associated gastrin concentrations in plasma with cardiovascular disease, but it was not pursued from which organ the gastrin originated [68]. We have examined the normal mammalian heart for gastrin expression without identifying transcriptional or translational products, e.g., the normal mammalian heart does not express the gastrin gene [25]. Rather, the related gut peptide and neurotransmitter cholecystokinin was identified in considerable concentrations. The CCK expression in porcine heart was found in both atrial and ventricular tissue, which suggests that even ventriculocytes may have a storage capacity, i.e., granules, for peptides (Figure 4) [69]. The peptides resulting from cardiac CCK gene expression were identified by purification and mass spectrometry. Interestingly, the cardiac proCCK processing differs from that of the gut with a major C-terminal form and a smaller corresponding N-terminal fragment (no current method of measurement exists for the latter). Taken together, cardiac CCK gene expression has been reported in both animal and cellular models, and the expression profile seems—at least to some degree—to associate to that of cardiac natriuretic peptides. Whether the cardio-specific processing leads to a true endocrine peptide remains to be established.

## 5. Conclusions

Studies on the endocrine heart have led to a paradigmatic change in our perception of mammalian cardiac function. Apart from the electrochemical and muscular aspects of cardiomyocytes, a capable and physiologically relevant production, maturation and release of peptide hormones has been documented. Studies on natriuretic peptides in the heart have shown that the post-translational phase of gene expression is not only relevant for human physiology but may prove implicated also in the development and, perhaps one day, cure of human cardiovascular disease.

## Figures and Tables

**Figure 1 biology-11-00971-f001:**
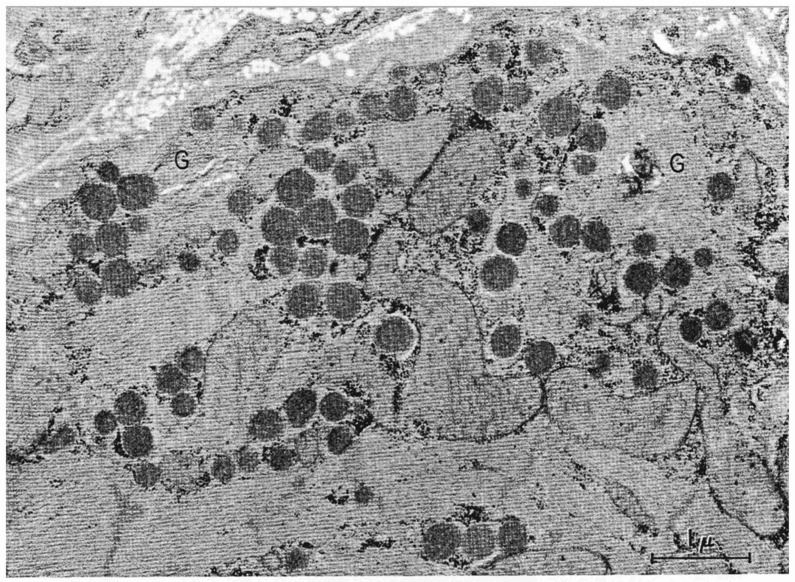
Granules (G) within atrial cardiomyocytes (taken from [12] with permission).

**Figure 2 biology-11-00971-f002:**
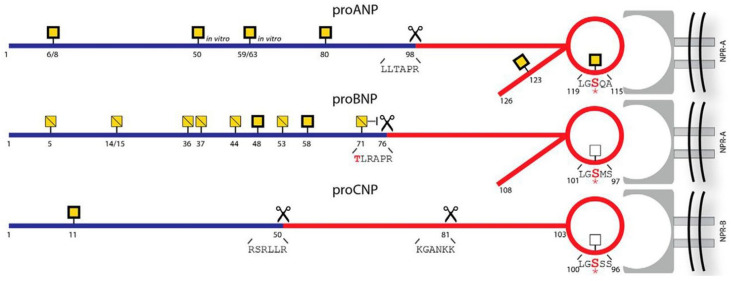
Natriuretic peptides and O-glycosylation (marked by yellow squares): Taken from [33] with permission.

**Figure 3 biology-11-00971-f003:**
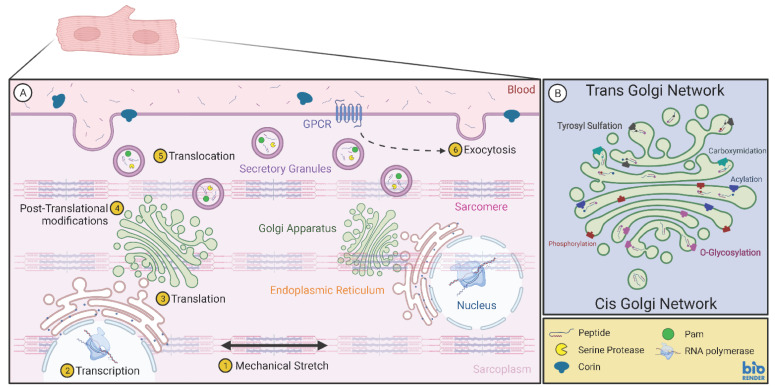
Schematic presentation of a cardiac myocyte containing a regulated secretory pathway (**A**) with its post-translational modification possibilities (**B**) (created in Biorender.com accessed on 17 May 2022).

**Figure 4 biology-11-00971-f004:**
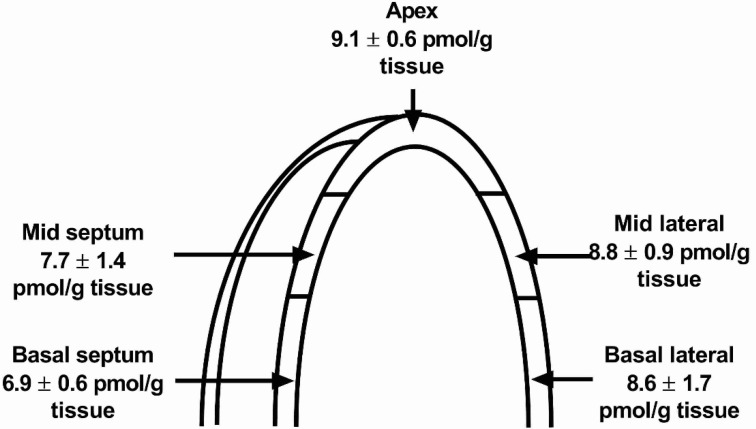
Ventricular expression of procholecystokinin. The different anatomical sections of the left ventricle represent sections normally visualized and evaluated by 2-dimensional echocardiography. Modified from [69].

## Data Availability

Not applicable.

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
