# Peer review of "Biochemistry of the Endocrine Heart"

_biology, 2022, doi:10.3390/biology11070971_

Round 1
Reviewer 1 Report
The manuscript is clear, interesting, well written, and contains relevant information.
I noticed several references very old. It would be necessary to update some of those references.
Are the authors under the process of asking for permission to use the figures?
Author Response
Reply to reviewers and editors,
Thank you for your expert comments on our mini review. I have chosen to address all comments in one mutual text; please find our comments and changes made to the manuscript in the below.
- A simple summary has been provided.
- Back matter completed.
- Several reviewers ask for permission to publish figures 1 and 2. This has been done, where permission for figure 1 can be forwarded. As for figure 2, there is no need for permission according to the Journal (JBC). We published that paper under open access and free rights to use the published material by others (if they refer to the original paper). The original paper is from our own group.
- The manuscript in the new setup has been proofread. Minor changes in the revised manuscript.
- One referee noted our optimism on ANP therapy in heart failure. As well noted by the referee, ANP as peptide therapy in its current form is not really a success and still only registered in Japan. We have modified the optimism to better reflect reality.
- Reviewer number 3 has several highly relevant suggestions for extending the mini review. For instance, addressing species differences would be very relevant to many readers. In addition, the regional differences in peptide and receptor expression, again in different species, would be valuable information for other researchers working with the natriuretic peptide system. However, these suggestions are worthy of full reviews on the specific subjects. The present review tries to give the readers a broad overview of several peptides produced and secreted by the heart rather than specific details on the natriuretic peptide system. We therefore hope that the reviewers and editors will allow us to keep the present broad focus and address the comments in forthcoming manuscripts.
- Reviewer number one suggests that we add code numbers for processing enzymes. I have not seen this done before in reviews. The enzyme names should be unique, as for instance the cardiac processing enzyme Corin.
- References: We have specifically chosen to refer to original literature wherever possible. Moreover, we prefer to refer to the earliest possible papers on the subjects, this to honor those who performed the pioneer work. We hope that you will allow us to keep this priority.
Reviewer 2 Report
The paper „Biochemistry of the Endocrine Heart” by Jens P. Goetze et al. describes in detail the mechanisms that lead to the production of natriuretic peptides and other peptides by cardiomyocytes. The main research question of this review article is the endocrine function of the myocardium. This review represents a well-written summary and is valuable compared to existing literature in giving an introduction about the pathophysiology of the “endocrine heart”. Unfortunately, no clear methodological details are given about the literature search. The conclusions are consistent with the evidence. References are appropriate. Figures are self-explanatory, but authors need to check whether they can use it without permission from figure authors.
In my opinion, the paper could be optimized by discussing already available treatment options for heart failure that modify the above-mentioned pathways (for example, neprilysin inhibitors).
Furthermore, the authors should discuss the distribution of active endocrine myocardial cells within the heart – are they evenly distributed throughout the atria or concentrated on one side or one location?
Author Response

(The authors gave the same response as above.)

Reviewer 3 Report
The authors describe the heart as a source of regulatory peptides. This area of research has been followed for decades and is not novel but efforts continue and might come useful for drug therapy of very frequent diseases.
Minor: please add the code numbers of the processing enzymes, otherwise the reader might misunderstand you statements
Major: please add a paragraph on species differences.;at least mouse, rat and human should be compared. Perhaps a table might help. What about lower species like avian preparations or frogs that are often used in basic research. ?
Moreover, from a reader´s perspective sequence information where you plot post translational modifications of ANF would be desirable.
I am missing at least for the human heart a comparison of regional data. There should be difference between left and right atrium and left and right ventricle because blood pressure is quite different and pressure is the signal for ANF release. and changes might occur with disease e.g. pulomonary hypertension.
A critical comparison which ANP derivatives can be measured in the clinic (e.g. pre pro ANF) and the current status of application would help.
I am missing data on atrial and ventricular arrhythmias and ANF level and secretion. Arrhythmias change cardiac pressures and ANP secretion would change.
There must be gene chips data on proteases for ANP and relaxin in the mouse heart and human heart in health and disease; please discuss.
Your view of ANP in heart failure is overly optimistic (reference 52). As far as this reviewer is aware all clinical studies outside Japan failed. Please clearly discuss this issue or delete the sentence
As concerns covalent modification of your peptides of interest why don t you discuss neovel mass spect possibilities and resultant data ?
Mention briefly that receptors for ANF exist. Are they also present in the myocyte it self that is intracellularly and do they function there ?.
In the heart there are many cell types. Is ANF found in smooth muscle cells, endothelial cells, fibroblasts and mast cells ? Does expression changes with diesease. ?
Are there any drugs that alter the expression of ANP processing enzymes in health or disease in animals or humans ?
Author Response

(The authors gave the same response as above.)
